# Distinct neural mechanisms underlie subjective and objective recollection and guide memory-based decision making

Yana Fandakova[1]*, Elliott G Johnson[2], Simona Ghetti[3]*

[1]Center for Lifespan Psychology, Max Planck Institute for Human Development, Berlin, Germany; [2]Human Development Graduate Group & Center for Mind and Brain, University of California at Davis, Davis, United States; [3]Department of Psychology & Center for Mind and Brain, University of California at Davis, Davis, United States

**Abstract** Accurate memories are often associated with vivid experiences of recollection. However, the neural mechanisms underlying subjective recollection and their unique role in decision making beyond accuracy have received limited attention. We dissociated subjective recollection from accuracy during a forced-choice task. Distractors corresponded either to non-studied exemplars of the targets (A-A' condition) or to non-studied exemplars of different studied items (A-B' condition). The A-A' condition resulted in higher accuracy and greater activation in the superior parietal lobe, whereas the A-B' condition resulted in higher subjective recollection and greater activation in the precuneus and retrosplenial regions, indicating a dissociation between objective and subjective memory. Activation in insular, cingulate, and lateral prefrontal regions was also associated with subjective recollection; however, during a subsequent decision phase, activation in these same regions was greater for discarded than for selected responses in anticipation of a social reward, underscoring their role in evaluating memory evidence flexibly based on current goals.

*For correspondence:
fandakova@mpib-berlin.mpg.de
(YF);
sghetti@ucdavis.edu (SG)

**Competing interests:** The authors declare that no competing interests exist.

## Introduction

The ability to remember past events in vivid detail is central to our experience owing to our proclivity to reflect or reminisce about our personal pasts. Recollection refers to the retrieval process yielding memories that capture the richness of our past, including such aspects as where, when, or with whom we experienced an event (*Yonelinas, 2002*). This process is often accompanied by the subjective feeling of vivid remembering, a sense of mental reliving the event (*Tulving, 1985*). At the neural level, recollection is associated with a network of brain regions including the medial temporal lobes, particularly the hippocampus and the parahippocampal gyrus, the retrosplenial/posterior cingulate cortex (PCC) and the precuneus, and the medial prefrontal cortex (PFC) and the angular gyrus (*Rugg and Vilberg, 2013*).

The subjective sense of vivid recollection is typically positively correlated with the accuracy of recollected details (*Moscovitch et al., 2016*). This subjective experience, therefore, represents a reliable cue for the degree to which a memory can be trusted to guide decision making. However, the investigation of the neural mechanisms supporting the subjective experience of recollection has received relatively little attention. This may be due to the noted strong correlation between accuracy and subjective experience, resulting in a convergence of these aspects within the dominant theories of recollection (e.g., *Bastin et al., 2019*; *Yonelinas, 2002*). Indeed, the neural network underlying recollection has been reliably identified across studies that require participants either to reflect on their subjective experience (*Daselaar et al., 2006*; *Vilberg and Rugg, 2007*) or to retrieve specific memory details (*Dobbins et al., 2002*; *Mitchell and Johnson, 2009*). However, accurate detail

retrieval, or objective recollection, does not necessarily align with the subjective experience of recollection.

Behavioral dissociations between accuracy and subjective recollection suggest that subjective assessments carry unique information that signal the diagnostic value of current mnemonic evidence. For example, after studying objects, faces, or parts of scenes, adults are more likely to correctly identify a studied item in a two alternative forced-choice (2AFC) task if the distractor corresponds to a non-studied exemplar of the target or a different section of the same scene (i.e., A-A' condition) than if the distractor corresponds to a non-studied exemplar of a different studied item (i.e., A-B' condition) (*Dobbins et al., 1998*; *Heathcote et al., 2009*; *Hembacher and Ghetti, 2017*; *Tulving, 1981*). However, participants are consistently less likely to claim subjective recollection in the A-A' condition, likely because it forces them to identify the most diagnostic feature from the comparison between the two perceptually similar probes. Despite gaining in memory accuracy, individuals may therefore recognize the challenge of identifying a diagnostic feature or may base their subjective recollection judgment only on that feature rather than on the entire experience (*Dobbins et al., 1998*; *Hembacher and Ghetti, 2017*). We refer to this reliance on specific details as 'specific retrieval' because it encourages participants to claim recollection only when they remember specific, highly diagnostic details. In contrast, the A-B' condition encourages a more global assessment of the identity of the presented probes. That is, in this condition, the presentation of the two dissimilar probes discourages identification of the most diagnostic features. Instead, it encourages participants to assess their retrieval of the probe as a whole, resulting in more errors, but also in a stronger sense of subjective recollection.

In the present study, we used this type of paradigm (*Figure 1*) to probe differences in the neural substrates of subjective recollection not confounded by memory accuracy. To confirm that the A-A' condition was associated with greater evidence accumulation as to be expected if recollection is based on remembering specific, highly diagnostic details, we modeled responses in the two experimental conditions using drift diffusion models (*Ratcliff, 1978*). We hypothesized that greater emphasis on diagnostic features in the A-A' condition would be manifested in greater evidence accumulation relative to the A-B' condition.

As for neural correlates, previous research offers a starting point for distinguishing subjective and objective recollection. Regions of the posterior parietal cortex (PPC) have been associated with retrieval of accurate details (*Wagner et al., 2005*) as well as with subjective recollection (*Chua et al., 2006*; *Simons et al., 2010*). Previous studies also hint at the possibility that subjective and objective recollection associated with the PPC are not always aligned. For example, patients with PPC damage have been shown to display equivalent source memory to matched controls, but considerably lower levels of subjective recollection (*Davidson et al., 2008Hower et al., 2014*). A separate line of research in aging populations has also identified dissociations between subjective and objective recollection. During a recognition memory task, Duarte and colleagues (*Duarte et al., 2008*) asked younger and older participants to indicate their subjective recollection, followed by a judgment about the temporal or spatial context in which items were studied. High-performing older adults showed similar levels of subjective recollection to younger adults but lower levels of objective recollection. This may be due to older adults retrieving comparable amounts of contextual details that were not diagnostic of the required source judgment, but that may have sufficed to induce a subjective feeling of recollection, similar to procedures encouraging retrieval of the target probe as a whole. Similarly, *Mark and Rugg, 1998* tested both subjective (remember-know procedure) and objective (source memory test) recollection in younger and older adults. While older adults displayed significantly lower objective memory, there were no age differences in subjective recollection. Even though these behavioral dissociations have been identified in some populations, mapping subjective and objective recollection to the corresponding neural circuits has been difficult due to the typically high correlation between the two. Corroborating previous research, we expected the functional dissociations between objective and subjective recollection to map onto regional differences in the PPC.

Functional dissociations between the ventral/medial and dorsal/lateral PPC have proved informative to distinguish objective and subjective aspects of retrieval (*Cabeza et al., 2008*; *Vilberg and Rugg, 2007*; *Wagner et al., 2005*). In a direct comparison of subjective (remember-know procedure) and objective recollection (source memory test), *Frithsen and Miller, 2014* showed enhanced activation in the angular gyrus in association with subjective recollection. Further dissociating the precision

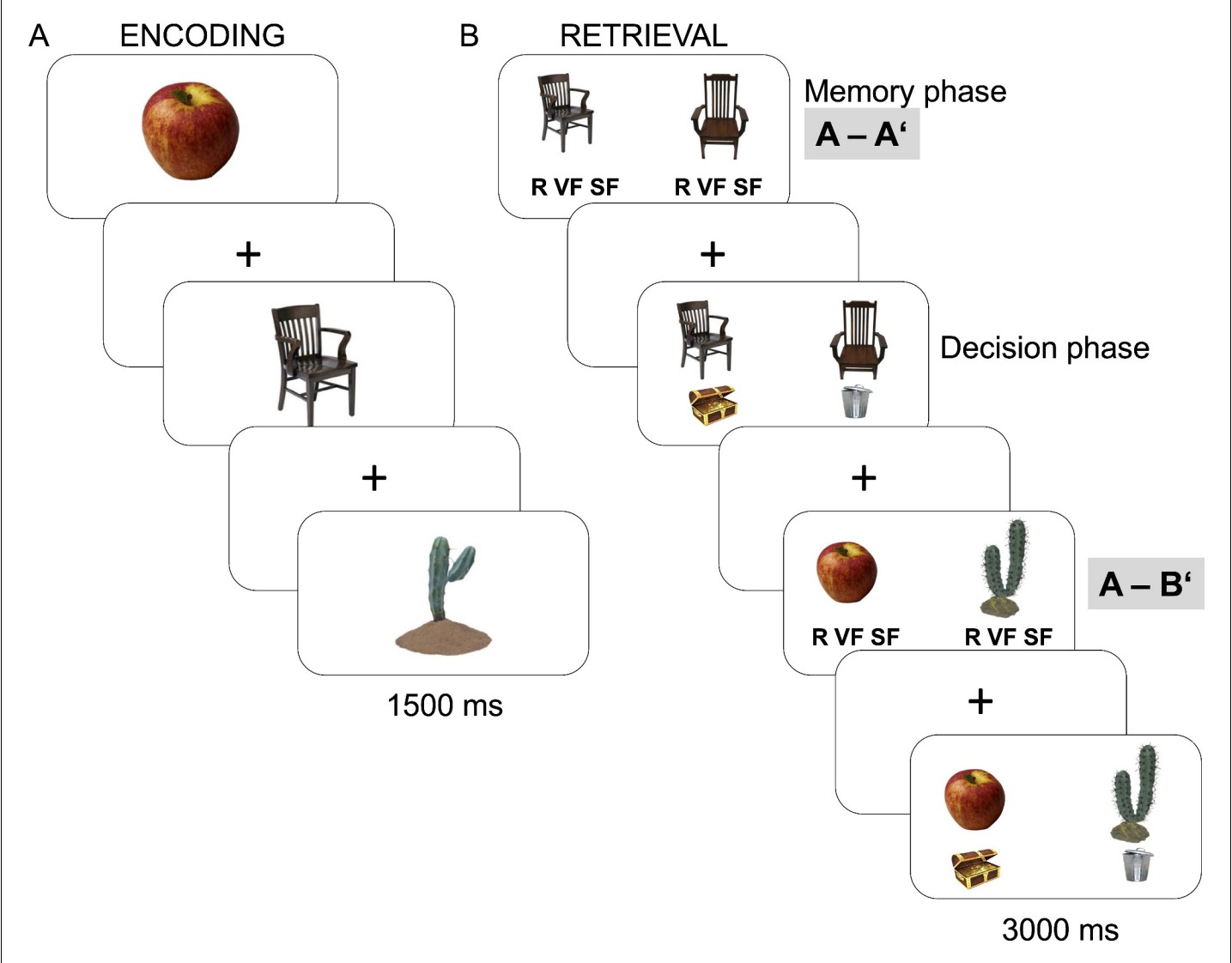

**Figure 1.** Experimental paradigm. (A) During encoding, participants incidentally encoded pictures of objects while making indoor-outdoor judgments with a jittered interstimulus interval (500–6500 ms). (B) During retrieval, participants completed a two alternative forced choice (2AFC) recognition test in which a studied target was presented either along with a distractor that was perceptually similar to the target (i.e., A-A' condition) or along with a distractor that was perceptually similar to another studied item (i.e., A-B' condition). Participants chose between a remember (R), very familiar (VF), or somewhat familiar (SF) judgment with the hand corresponding to the position of the image they recognized as the target. Target and distractor positions were randomized across trials. After a jittered fixation period (500–6500 ms) following the memory phase, the two alternatives were presented again and participants were asked to select their memory to be counted toward their final score by pressing the button corresponding to the treasure chest or to discard the item by pressing the button corresponding to the trash can (i.e., decision phase). The position of the treasure chest and trash can buttons were randomized across trials.

of memory retrieval from memory vividness, *Richter et al., 2016* reported that activation in the angular gyrus was associated with objective memory precision, whereas activation in the precuneus was associated with subjective memory vividness. Of particular relevance, Favila and colleagues (*Favila et al., 2018*) examined differences between ventral and dorsal PPC with respect to how the representation of object features in these regions is influenced by retrieval goals. Participants were presented with stimuli that varied across color and object category, but these features were tested separately, allowing evaluation of the degree to which PPC regions carried information about goal-relevant vs. goal-irrelevant feature information. The results demonstrated that memory goals biased feature representation toward relevant information in dorsal PPC but not in ventral PPC. Based on

these reported dissociations, we expected a dissociation between dorsal PPC and precuneus activation. We predicted enhanced dorsal PPC activation in the A-A' relative to the A-B' condition because the former encourages identification of more precise, diagnostic features. This would corroborate previous research showing that this region is under stronger influence from top-down control and retrieval goals (*Cabeza et al., 2008*). In contrast, we predicted enhanced ventral PPC activation, including the angular gyrus and/or the precuneus, in the A-B' condition because this condition triggers a more global assessment of the test probes, resulting in greater subjective recollection (*Hembacher and Ghetti, 2017*).

Subjective assessments of recollection are metacognitive acts (*Koriat and Goldsmith, 1996*) and as such can be expected to be recapitulated in regions associated with metacognitive monitoring and evaluation (*Vaccaro and Fleming, 2018*). Metacognitive assessments of decision confidence, which represent retrospective metacognitive monitoring, have been associated with activation in dorsal anterior cingulate cortex (dACC) and anterior insula supporting performance or error monitoring (*Fandakova et al., 2018*), and with activation in lateral PFC to support the evaluation of retrieved content in the context of current task demands (*Bang and Fleming, 2018*). Interestingly, a behavioral study found that participants were more likely to bet on their responses in the A-B' condition in anticipation of a social reward (*Hembacher and Ghetti, 2017*), suggesting that cues utilized to support subjective recollection are more heavily factored in subsequent decision making than cues associated with accurate response selection. Thus, we set to examine the neural mechanisms that connect subjective recollection with decision making. We hypothesized that when participants are asked to select a memory, this decision would reflect their assessment of how well they had performed on a given trial in the context of the specific task goal, such as obtaining a higher performance score (*Koriat and Goldsmith, 1996*). Thus, we expected neural activation in regions associated with metacognitive monitoring (i.e., anterior insula and dACC) and evaluation (i.e., lateral PFC) to be aligned to participants' selection decisions at the time of making those decisions. Taken together, we expected that our task would engage fronto-parietal regions, with parietal regions tracking the dissociation between objective and subjective recollection during the memory phase, and insular, cingulate, and lateral PFC regions tracking goal-relevant dimensions of the task associated with subsequent decision making.

## Results

### Behavioral results

#### Dissociation between memory accuracy and subjective recollection

We examined differences in memory accuracy and subjective recollection as a function of experimental condition. First, we used mixed-effects logistic regressions to predict accuracy as a function of experimental condition. Consistent with previous studies, memory accuracy was higher in the A-A' than in the A-B' condition, estimated difference (Est.) = 0.38, SE = 0.05, p<0.0001 (*Figure 2A*). Next, we examined whether the probability for a remember judgment (as opposed to a familiar judgment) differed as a function of experimental condition, accuracy, and their interaction. As predicted, there was an effect of experimental condition, Est. = 1.35, SE = 0.11, p<0.0001, with a higher rate of remember responses in the A-B' than in the A-A' condition. The interaction between experimental condition and accuracy was also significant, Est. = −0.64, SE = 0.13, p<0.0001, suggesting that although remember judgments were overall more likely in the A-B' condition, the difference as compared with the A-A' condition was more pronounced for incorrect responses (*Figure 2B*). This interaction was found above and beyond the expected main effect of accuracy showing that remember responses were overall more likely for accurate than for inaccurate responses, Est. = 0.96, SE = 0.10, p<0.0001 (*Figure 2B*).

Together, these results indicate that memory accuracy and subjective recollection were behaviorally dissociated across experimental conditions: Accuracy was higher in the A-A' condition, but subjective recollection was more frequent in the A-B' condition.

#### Decision making aligns with subjective recollection

Next, we sought to confirm that subjective recollection drives the decision to select responses toward participants' final score. Mixed-effect logistic regressions were used to predict the

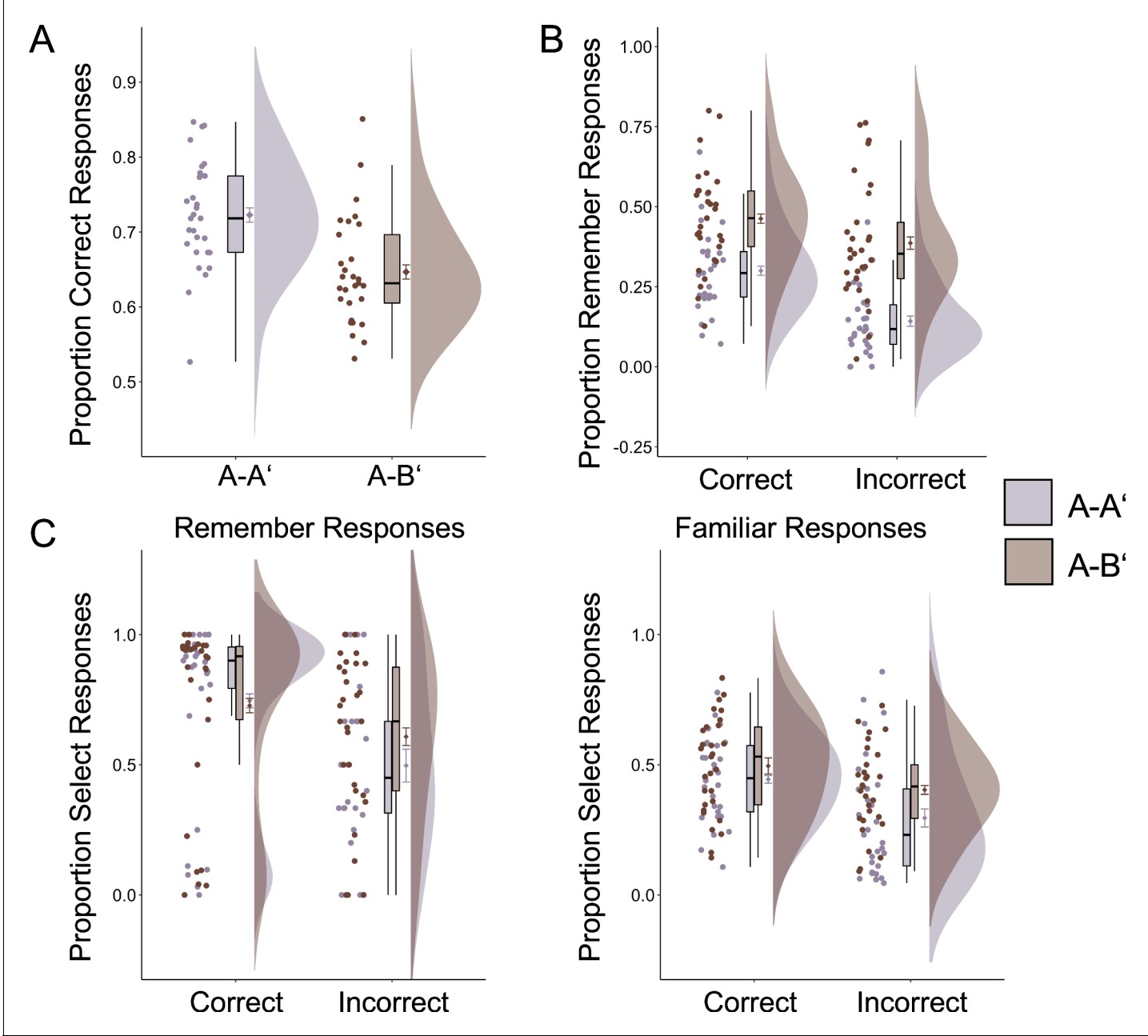

**Figure 2.** Behavioral results. (**A**) Accuracy across experimental conditions. There were more correct responses in the A-A' condition. (**B**) Proportion of remember judgments out of all correct and incorrect responses across experimental conditions. Participants more often claimed subjective recollection in the A-B' condition both for accurate and inaccurate responses. (**C**) Proportion of memory responses that participants selected to count toward their final score split as a function of experimental condition, accuracy, and subjective judgment. For correct responses, there were no condition differences in selection rates for remember responses, but participants' tendency to select familiar responses increased in the A-B' condition. Error bars around condition means represent standard error of the mean. For plots of estimated logistic regression functions, see *Figure 2—figure supplement 1*. The online version of this article includes the following source data and figure supplement(s) for figure 2:

**Source data 1.** Accuracy across experimental conditions.
**Source data 2.** Proportion of remember responses by experimental condition and accuracy.
**Source data 3.** Proportion of select responses by experimental condition, accuracy, and subjective judgment.
**Figure supplement 1.** Estimated results from logistic regressions on accuracy, remember judgments, and select decisions.

probability of selecting a memory as a function of experimental condition (A-A' vs. A-B'), subjective rating (remember vs. familiar), memory accuracy, and their interactions. Participants were more likely to select an answer to count toward their final score in the A-B' than in the A-A' condition, Est. = 0.72, SE = 0.12, p<0.0001, above and beyond the greater tendency to select memories associated with remember relative to familiar judgments, Est. = 1.64, SE = 0.20, p<0.0001, and with accurate compared to inaccurate responses, Est. = 1.02, SE = 0.09, p<0.0001. Critically, we also observed a three-way interaction between condition, rating, and accuracy, Est. = −0.78, SE = 0.30, p=0.02 (*Figure 2C*).

For correct responses, selecting a response was more likely in the A-B' than in the A-A' condition, Est. = 0.45, SE = 0.08, p<0.0001, and more frequent for remember than for familiar judgments, Est. = 2.69, SE = 0.14, p<0.0001. There was also a significant condition by subjective judgment interaction, Est. = −0.64, SE = 0.18, p=0.0004, such that selection rates were similar for remember ratings between conditions, but participants extended their tendency to select A-B' trials even to some familiar judgments.

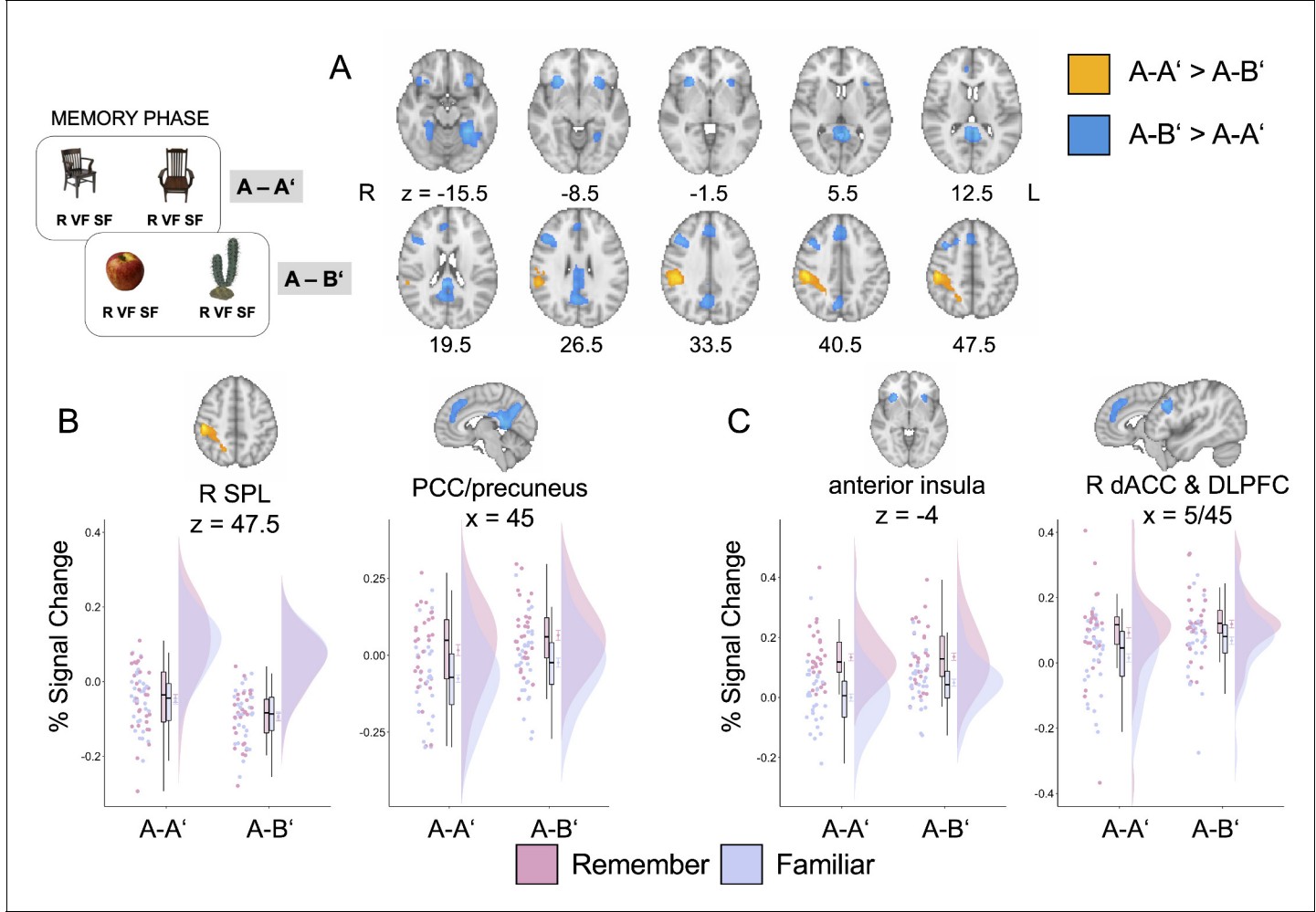

**Figure 3.** Neuroimaging results for the memory phase. (**A**) Results of a whole-brain comparison between experimental conditions. (**B**) Differences between remember and familiar judgments across conditions in parietal areas identified in A. (**C**) Differences between remember and familiar judgments across conditions for regions identified in A that have been implicated in metacognitive monitoring and appraisal. Error bars around condition means represent standard error of the mean. R SPL = right superior parietal lobe; PCC = posterior cingulate cortex, dACC = dorsal anterior cingulate; R DLPFC = right dorsolateral prefrontal cortex.

The online version of this article includes the following source data and figure supplement(s) for figure 3:

**Source data 1.** Signal change estimates plotted in *Figure 3* by region of interest (ROI) and contrast.
**Figure supplement 1.** Whole-brain comparisons of remember judgments relative to familiar judgments in the A-A' condition and in the A-B' condition.

For incorrect responses, selection rates were again higher in the A-B' than in the A-A' condition, Est. = 0.74, SE = 0.12, p<0.0001 and more frequent for remember than for familiar judgments, Est. = 1.56, SE = 0.20, p<0.0001; the interaction between subjective judgment and experimental condition was not significant, Est. = 0.08, SE = 0.24, p=0.73 (*Figure 1C*). Together, these results indicate that participants were more likely to select an answer toward their final score in the A-B' than in the A-A' condition, particularly in association with remember responses. For accurate responses, participants also showed a higher likelihood to select familiar responses in the A-B' condition.

## Faster evidence accumulation occurs in the A-A' condition

If highly similar lure items in the A-A' condition induce participants to engage in examination of specific diagnostic features, then the A-A' condition should be associated with a higher rate of evidence accumulation compared to the A-B' condition. We thus used drift diffusion models (*Ratcliff, 1978*) to compare drift rates and threshold separation between conditions. An ANOVA with condition (A-A' vs. A-B') as a within-subject factor revealed significant differences in drift rate, $F(1,28) = 15.48$, p=0.002, $\eta_p^2 = 0.36$, with higher drift rate estimates in the A-A' condition, M = 2.06, SD = 0.62, than in the A-B' condition, M = 1.22, SD = 0.61. Threshold separation, reflecting how cautious participants were in their memory choice, did not differ between conditions, $F(1,28) = 3.59$, p=0.07, $\eta_p^2 = 0.11$ (A-A': M = 2.42, SD = 0.25; A-B': M = 2.51, SD = 0.21). Together, evidence accumulation followed the pattern found for accuracy differences between conditions, such that participants showed faster evidence accumulation in the A-A' than in the A-B' condition, consistent with the idea that the A-A' condition promotes specific retrieval of more diagnostic features.

## Neuroimaging results: memory phase

### Activations in parietal regions dissociate subjective and objective recollection

We performed whole-brain analyses to examine differences between the A-A' and A-B' conditions during the memory phase. Here, we were particularly interested in PPC regions, in which we expected a dissociation aligned to condition with stronger engagement of dorsal PPC in the A-A' condition, but stronger engagement of medial/ventral parietal regions in the A-B' condition. Consistent with our expectations, the contrast A-A' > A-B' revealed greater activation in a cluster encompassing the right superior parietal lobe (SPL) and the supramarginal gyrus (*Figure 3A*). Instead, the contrast A-B' > A-A' revealed greater activation in the bilateral retrosplenial cortex, PCC, and the precuneus (*Figure 3A*).

Given that behavioral rates of subjective recollection differed between experimental conditions, we next sought to evaluate whether activation in these parietal clusters varied as a function of subjective assessments, namely remember vs. familiar judgments. To this end, we conducted a mixed ANOVA on the parameter estimates with condition (A-A' vs. A-B') and subjective judgment (remember vs. familiar) as within-subject factors. Given that the regions of interest (ROIs) were identified as showing a main effect of condition, we focused only on the main effect of subjective judgment and the interaction between condition and subjective judgment here. In the SPL, which showed higher activation in the A-A' condition, there were neither differences between remember and familiar judgments, $F(1,28) = 0.03$, p=0.86, $\eta_p^2 = 0.001$, nor an interaction between subjective judgment and experimental condition, $F(1,28) = 0.60$, p=0.45, $\eta_p^2 = 0.021$ (*Figure 3B*). In contrast, in the PCC/precuneus cluster, which showed higher activation in the A-B' condition, activation was enhanced for remember relative to familiar judgments across both experimental conditions, $F(1,28) = 30.54$, p<0.001, $p_{adj} < 0.001$, $\eta_p^2 = 0.52$ (*Figure 3B*). We observed no significant interaction between subjective judgment and experimental condition in this region, $F(1,28) = 0.02$, p=0.90, $p_{adj} = 0.90$, $\eta_p^2 = 0.001$. Additional analyses comparing remember to familiar judgments in each experimental condition demonstrated that the left angular gyrus showed greater activation for remember than for familiar judgments across both experimental conditions (see *Figure 3—figure supplement 1*).

Taken together, there was a clear dissociation between experimental conditions in the parietal cortex. The SPL showed greater engagement in the A-A' condition, presumably reflecting the increased scrutiny promoted by this condition. In contrast, the PCC/precuneus was more strongly

engaged in the A-B' condition, and consistent with behavioral trends, showed greater activation for subjective recollection.

## Frontal and cingulo-opercular regions track dimensions that are relevant for decision making

The contrast between A-B' and A-A' also revealed increased activation in the bilateral anterior insula, the dACC, and the right dorsolateral PFC (dlPFC) (*Figure 3A*), regions that have been implicated in retrospective metacognitive assessments. To examine the role of these regions in subjective judgments, we again conducted a mixed ANOVA on the parameter estimates with condition (A-A' vs. A-B') and subjective judgment (remember vs. familiar) as within-subject factors. The bilateral anterior insula showed enhanced activation for remember relative to familiar judgments across conditions, $F(1,28) = 73.43$, $p<0.001$, $\eta_p^2 = 0.72$. This main effect was accompanied by an interaction with experimental condition, $F(1,28) = 6.19$, $p=0.02$, $p_{adj} = 0.057$, $\eta_p^2 = 0.18$ (*Figure 3C*). Post hoc tests indicated that although anterior insula activation was similar across conditions for remember judgments, $t(28) = 0.12$, $p=0.91$, it was enhanced in the A-B' condition for familiar judgments, $t(28) = 3.18$, $p=0.004$. In addition, the cluster encompassing the dACC and dlPFC showed more pronounced engagement in remember than in familiar judgments across both experimental conditions, $F(1,28) = 29.49$, $p<0.001$, $p_{adj} <0.001$, $\eta_p^2 = 0.51$, but no interaction with condition, $F(1,28) = 0.900$, $p=0.35$, $p_{adj} = 0.53$, $\eta_p^2 = 0.03$ (*Figure 3C*).

Taken together, these results suggest that anterior insular, cingulate, and dlPFC regions track subjective recollection, in line with their role in metacognitive assessments, and in line with the tendency to select memories to submit for performance ranking among remembered answers. At the same time, these regions were enhanced in the A-B' condition suggesting that these areas may be ideally suited to carry over metacognitive information about subjective assessments associated with different contexts to the decision phase.

## Hippocampal activations respond to memory accuracy

We tested whether activation in an anatomical mask of the bilateral hippocampus varied by experimental condition (A-A' vs. A-B') and accuracy (correct vs. incorrect). The results revealed a main effect of accuracy, $F(1,28) = 6.39$, $p=0.02$, $\eta_p^2 = 0.19$, with enhanced activation for correct responses. There was also a main effect of condition, $F(1,28) = 9.17$, $p=0.01$, $\eta_p^2 = 0.25$ with enhanced activation in the A-B' condition. We found no evidence for an accuracy-by-experimental condition interaction, $F(1,28) = 0.04$, $p=0.85$, $\eta_p^2 = 0.001$, suggesting similar retrieval success effects across experimental conditions in the hippocampus.

## Neuroimaging results: decision phase

### Frontal and cingulo-opercular regions track betting decisions

To assess whether patterns of activation during memory retrieval persisted during decision making, we took several complementary approaches. First, we examined decision-related activation in the parietal regions that differentiated between conditions during the preceding memory phase. Thus, we performed mixed ANOVAs with condition (A-A' vs. A-B'), decision (select vs. discard), and their interaction. In the SPL, which showed enhanced activation in the A-A' condition during the memory phase, we again observed enhanced activation for the A-A' compared to the A-B' condition during the decision phase, $F(1,28) = 9.29$, $p=0.01$, $\eta_p^2 = 0.25$. SPL activation was not modulated by the decision to select or discard the memory, $F(1,28) = 0.99$, $p=0.33$, $\eta_p^2 = 0.03$, and the interaction between condition and betting decision was not significant, $F(1,28) = 0.30$, $p=0.59$, $\eta_p^2 = 0.01$ (*Figure 4A*). In the PCC/precuneus cluster, which showed enhanced activation in the A-B' condition during the memory phase, we observed a trend for a main effect of condition with greater engagement in the A-A' condition, $F(1,28) = 5.190$, $p=0.031$, $p_{adj} = 0.09$, $\eta_p^2 = 0.156$, no significant main effect of decision to select or discard a response, $F(1,28) = 1.62$, $p=0.21$, $p_{adj} = 0.21$, $\eta_p^2 = 0.06$, and no decision by experimental condition interaction, $F(1,28) = 0.011$, $p=0.92$, $p_{adj} = 0.92$, $\eta_p^2 = 0.00$ (*Figure 4A*). Thus, these analyses did not reveal any decision-based modulation in the parietal regions identified during the memory phase.

Second, data analysis during the memory phase also revealed increased activation in the bilateral anterior insula and a cluster including dACC and dlPFC regions, which are consistently implicated in

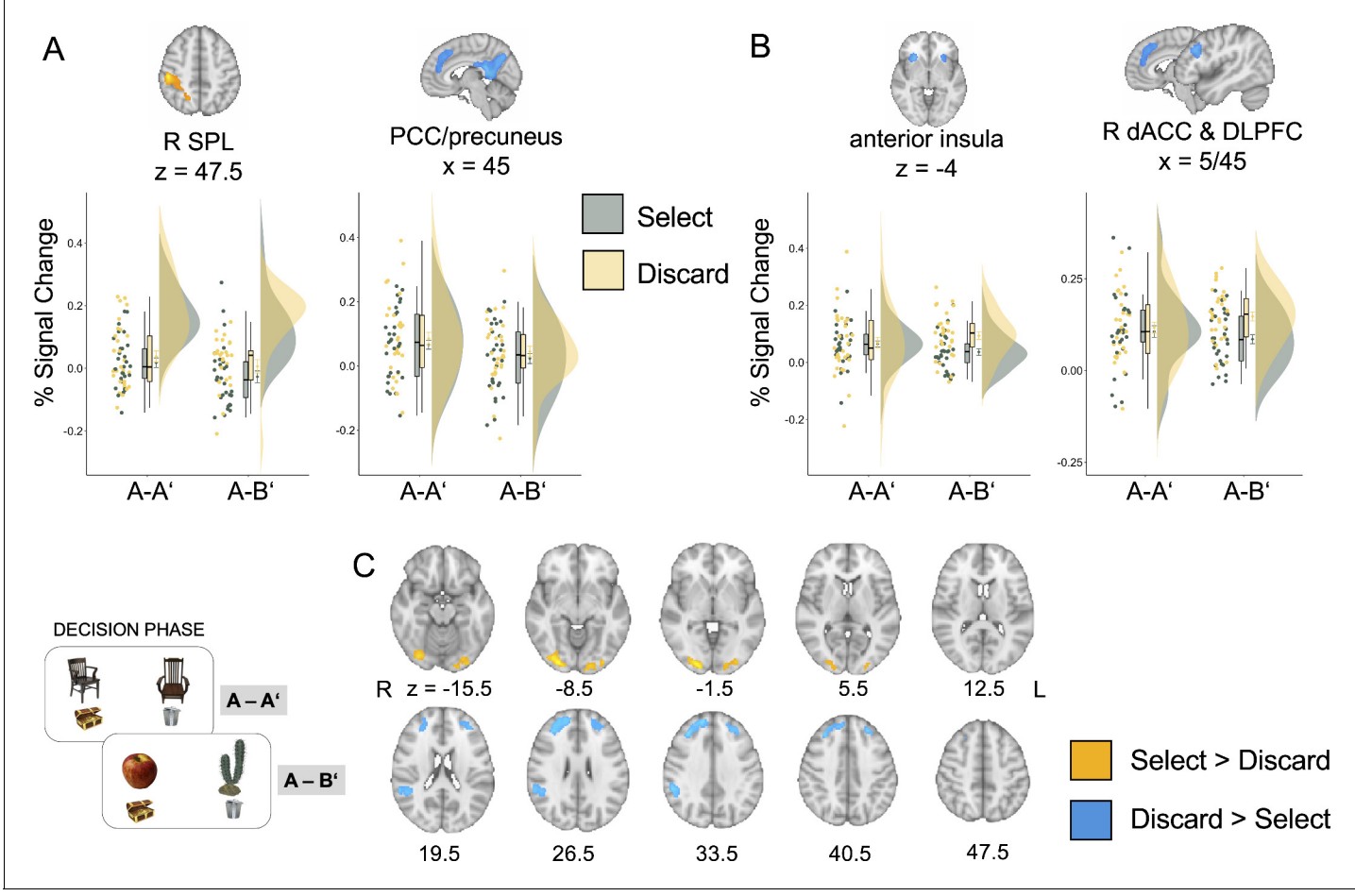

**Figure 4.** Neuroimaging results for the decision phase. (A) Differences between select and discard decisions across conditions in the PPC areas identified during the memory phase. (B) Differences between select and discard decisions across conditions for the regions identified during the memory phase that have been implicated in metacognitive monitoring and appraisal. (C) Results of a whole-brain comparison comparing select and discard decisions. Error bars around condition means represent standard error of the mean. R SPL = right superior parietal lobe; PCC = posterior cingulate cortex, dACC = dorsal anterior cingulate; R DLPFC = right dorsolateral prefrontal cortex.
The online version of this article includes the following source data for figure 4:

**Source data 1.** Signal change estimates plotted in *Figure 4* by region of interest (ROI) and contrast.

monitoring and control of memory retrieval. These regions showed enhanced activation in the A-B' condition relative to the A-A' condition with varying engagement as a function of subjective assessment. They are therefore excellent candidates to carry over metacognitive information about subjective assessments from the memory to the decision phase. Thus, we expected activation in these regions to be modulated by decisions to select or discard responses. In line with this expectation, in the bilateral anterior insula, there was a main effect of decision, $F(1,28) = 4.81$, $p=0.04$, $p_{adj} = 0.06$, $\eta_p^2 = 0.15$, along with a decision-by-confidence interaction, $F(1,28) = 5.88$, $p=0.02$, $p_{adj} = 0.03$, $\eta_p^2 = 0.17$ (*Figure 4B*). There was no main effect of experimental condition, $F(1,28) = 0.02$, $p=0.90$, $p_{adj} = 0.90$, $\eta_p^2 = 0.001$. A similar pattern emerged in the cluster encompassing right dACC and dlPFC, with a main effect of decision to select, $F(1,28) = 8.65$, $p=0.01$, $p_{adj} = 0.03$, $\eta_p^2 = 0.236$, as well as an experimental condition by decision interaction, $F(1,28) = 6.09$, $p=0.02$, $p_{adj} = 0.03$, $\eta_p^2 = 0.18$ (*Figure 4B*). Again, there was no main effect of experimental condition, $F(1,28) = 0.07$, $p=0.79$, $p_{adj} = 0.90$, $\eta_p^2 = 0.003$. Across all areas, activation was enhanced for decisions to discard a response, especially in the A-B' condition (*Figure 4B*). Thus, although these regions were identified through our experimental manipulation in the memory phase, the effect of the experimental

manipulation was no longer dominant during the decision phase, whereas the effect of decision to select or discard emerged.

Finally, we sought to identify regions that were sensitive to memory decisions, by comparing activation for decisions to select or discard a memory using whole-brain analysis. In analogy to the ROI results above, frontal regions showed enhanced activation for the decision to discard a memory, including bilateral middle frontal gyrus extending into the bilateral frontal pole, and the right angular gyrus (*Figure 4C*). In contrast, the decision to select a response was associated with increased activation in ventral temporal areas and posterior hippocampus (*Figure 4C*).

Taken together, parietal regions that dissociated between subjective and objective recollection as a function of our experimental manipulation during the memory phase demonstrated overall condition differences during decision making. In contrast, the anterior insula, dACC, and dlPFC showed distinct activation profiles during the memory phase and the decision phase, such that their activation was enhanced when participants discarded a response, especially in the A-B' condition.

## Discussion

The goal of the present study was to examine the neural basis underlying subjective recollection and subsequent decision making. To this end, we used an experimental paradigm that allowed us to dissociate subjective recollection from memory accuracy. When participants encountered a target studied item along with a perceptually similar (and semantically identical) lure, their memory accuracy was higher, but their subjective recollection was lower (A-A' condition) compared to when participants encountered a target studied item along with a lure that was similar to another studied item (A-B' condition). At the neural level, we observed a dissociation within the PPC during the memory phase of the experiment. Specifically, the A-A' condition was associated with enhanced activation in the right SPL across both remember and familiar responses. In contrast, the A-B' condition was associated with enhanced activation in the bilateral precuneus and retrosplenial cortex, which also showed overall greater engagement during remember than during familiar responses in both experimental conditions. Together, these findings suggest that when the retrieval context favors identifying more precise diagnostic features, the resulting higher accuracy is associated with dorsal PPC at the expense of subjective recollection. In contrast, activation in the precuneus and retrosplenial cortex tracks the experience of subjective recollection, which is promoted in a retrieval context that favors global retrieval.

The PPC has a long-established role in memory retrieval (*Wagner et al., 2005*), but its precise role is a matter of ongoing debate (*Cabeza et al., 2008*; *Gilmore et al., 2015*). It has been implicated as one of the core recollection regions (*Gilmore et al., 2015*; *Rugg and Vilberg, 2013*) as well as in the subjective experience of recollection (*Simons et al., 2010*). However, subjective recollection and the retrieval of accurate details usually go hand in hand, making it difficult to isolate the unique neural underpinnings of the phenomenological experience that lies at the heart of episodic memory. At the same time, there is evidence that subjective and objective memory measures can be dissociated (e.g., *Harlow and Yonelinas, 2016*). For example, older adults give similar or even higher judgments of subjective recollection compared to younger adults, yet they show considerable deficits in measures of objective recollection (*Addis et al., 2011*; *Duarte et al., 2008*; *Folville et al., 2020*; *Mark and Rugg, 1998*). Dissociations between subjective and objective memory measures are also observed in neuropsychiatric disorders, such as schizophrenia (*Huron et al., 1995*). At the neural level, a meta-analysis comparing regions implicated in subjective vs. objective recollection *Spaniol et al., 2009* found that prefrontal areas showed greater engagement in objective recollection, whereas parietal, hippocampal, and parahippocampal areas were more strongly associated with subjective recollection. However, objective and subjective recollections were highly correlated in most of these studies, precluding clear conclusions regarding the neural underpinnings of subjective recollection.

The experimental dissociation between SPL and precuneus/retrosplenial cortex during the memory phase is in line with the attention-to-memory framework (*Cabeza et al., 2008*) and with recent evidence implicating precuneus in memory vividness (*Richter et al., 2016*). The A-A' condition is more likely to encourage the identification of the most diagnostic features from the comparison between the two similar probes (*Dobbins et al., 1998*). Consistent with this, we found that the A-A' condition was associated with higher rates of evidence accumulation. Together, our behavioral and

neural results converge to suggest that the SPL plays an important role in supporting diagnostic processing during retrieval. These findings are in line with the established role of the SPL in the dorsal attention network (*Corbetta and Shulman, 2002*) and corroborate a recent suggestion that this region supports higher memory accuracy via its role in perceptual search and attention to probes during retrieval (*Sestieri et al., 2017*).

The precuneus showed a different activation profile in our task. In line with previous evidence of the involvement of this area in subjective recollection (*Richter et al., 2016*), it showed greater activation for remember relative to familiar responses across conditions. Of note, the present results additionally showed overall greater activation in the precuneus in the A-B' condition, suggesting that this region is modulated by the current retrieval context and can be promoted when a more global assessment of the probe is encouraged. The precuneus has been implicated consistently in autobiographical retrieval (*McDermott et al., 2009*) and may be one of the distinguishing features of individuals with superior memory (*Mazzoni et al., 2019*). A recent study compared retrieval of recently studied events and autobiographical events (*Chen et al., 2017*) and found that while memory for recently learned scenes was associated with activation of the posterior middle intraparietal sulcus, autobiographical memories involving scenes engaged the retrosplenial cortex and the precuneus. Autobiographical retrieval was also more likely to evoke subjective recollection, similar to our A-B' condition. This idea is in accordance with the suggestion that the diagnostic value of the recollective experience varies depending on retrieval context (*Dobbins et al., 1998*). In line with our behavioral findings, our neural findings reveal a dissociation across parietal regions, with SPL activation tracking the retrieval of diagnostic features in the A-A' condition, and precuneus activation tracking the subjective experience of recollection. Our results are consistent with clinical studies implicating ventral PPC in subjective recollection. One study showed that two patients with inferior parietal lesions (extending into the precuneus in one of the two) exhibited similar levels of recognition memory as did controls but were less likely to report high confidence for correctly recognized items (*Hower et al., 2014*). Similarly, *Simons et al., 2010* showed that patients with bilateral parietal lesions performed as well as matched controls on source memory but exhibited reduced confidence in their source judgments, again linking PPC lesions to impairments in the experience of rich and vivid episodic details typically associated with recollection judgments. Finally, Ciaramelli and colleagues (2017) showed that subjective recollection experiences in healthy controls were marked by multi-featural context retrieval, that is, participants remembered both the position and the color of items that they recognized as old. In contrast, subjective recollections in PPC patients were less likely to involve multi-featural retrieval, indicating a misalignment between subjective and objective recollection following PPC damage. These results may be interpreted in a number of ways. On the one hand, patients may truly experience reduced richness and vividness of remembered details, due perhaps to problems with mental imagery. On the other hand, it is also possible that patients' deficits are related to difficulties with metacognitive evaluation of memory evidence. By dissociating subjective and objective recollection experimentally and requiring memory selections in anticipation of a social reward, the present study reveals the close interplay between the subjective experience of recollection and metacognitive monitoring, and their role for subsequent decision making.

The present results also contribute to a growing body of literature examining the contributions of PPC to different aspects of episodic recollection. Using multivoxel pattern analysis, *Kuhl and Chun, 2014* showed that the angular gyrus is not only involved in vivid recollections but also carries information about the stimulus category and event-specific information. Notably, these effects were reduced or absent in the SPL, in line with a dissociation between these regions. Angular gyrus activation has also been reported during false memory, that is, when participants experience novel items as studied (*McDermott et al., 2017*), reinforcing the idea that this area is associated with the processing of the subjective feeling of remembering. Accordingly, a recent account of the role of the angular gyrus in episodic memory suggested that it is involved in contextual integration (*Ramanan et al., 2018*). More specifically, this account suggests that sensory-perceptual and emotionally salient features are integrated in the angular gyrus and form the basis for the subjective experience of vivid recollection. In line with this account and previous studies, the left angular gyrus showed enhanced activation for remember judgments across both experimental conditions. However, we found that increased subjective recollection in the A-B' condition was associated with the precuneus instead. Previous studies in which accurate detail retrieval and subjective recollection were examined separately also found these two aspects of episodic memory to be associated with

the angular gyrus and the precuneus, respectively (*Richter et al., 2016*). Of note, in our study, the precuneus showed an overall higher activation in the A-B' condition and also differentiated remember from familiar responses in the A-A' condition. Overall, the present results suggest that the precuneus may be involved in enhancing the salience of retrieved memories in line with its role in mental imagery (*Cavanna and Trimble, 2006*).

As compared to the A-A' condition, the A-B' condition also triggered the overall greater activation in the bilateral anterior insula, dACC, and right dlPFC regions during the memory phase. In addition, these regions were more active for remember judgments across both experimental conditions. These results fit well with previous meta-analyses that have implicated the anterior insula in association with the salience of the retrieved output (*Kim, 2010*). Specifically, the anterior insula may signal salient information during retrieval in light of concurrent goals (*Craig, 2009*; *Fandakova et al., 2018*). From this perspective, we speculate that the enhanced anterior insular activation in the A-B' condition (and in other tasks emphasizing judgments of subjective recollection) reflects the fact that the most goal-salient aspect of the task is to identify which items are subjectively recollected.

In separate literature on metacognition, the anterior insula as well as the dACC have been implicated in retrospective metacognitive monitoring (*Fleck et al., 2006*; *Morales et al., 2018*). Furthermore, lateral PFC regions have been implicated in the evaluation of retrieval outputs in context of current goals and demands (*Chua et al., 2006*; *Fandakova et al., 2014*; *Kim and Cabeza, 2009*), and in making decisions based on retrieved output (*Fandakova et al., 2018*). For example, *Fandakova et al., 2018* showed that the anterior insula and dACC were activated both when participants inaccurately remembered the context in which an object was encountered as well as when they decided to withhold their reports of memory for item–context associations, whereas the anterior PFC was uniquely engaged when participants withheld an answer. The increased engagement of these regions during retrieval in the A-B' condition is consistent with the idea that this condition poses greater demands on retrospective metacognitive monitoring and control due to a greater tendency to claim recollection despite less mnemonic evidence.

Furthermore, according to metacognitive frameworks, decisions and actions based on memory result from a metacognitive appraisal process (*Koriat and Goldsmith, 1996*). Thus, the activation profiles of these brain regions, which varied both as function of experimental condition and as function of subjective judgment, are particularly well suited to carry over information about retrieval output from the memory to the decision phase. Consistent with the idea that retrospective metacognitive monitoring processes mediate the effects of subjective recollection on subsequent decision making, the ROIs in the bilateral anterior insula, dACC, and dlPFC found during the memory phase showed enhanced activation when participants decided to discard a response, especially in the A-B' condition. Thus, it is clear that these regions do not respond to simple variations in memory strength, be it objective or subjective. If this were the case, we would have expected increased activation for remember responses during the memory phase and increased activation of selected memories during the decision phase. Instead, during the memory phase, these regions tracked whether memories were claimed to be recollected, given the relevance of subjective recollection for the upcoming decision. During decisions, however, they were engaged more in eliminating the memories that should not be submitted to count toward participants' final score, the socially motivated reward used in the present study. These results were further supported by the direct comparison of select and discard responses during decision making, which confirmed that lateral PFC is involved in decisions to discard a memory.

It should be noted that while participants were instructed to select about half of their responses, they placed more than 50% of their answers into the treasure chest, possibly leading to greater demands on monitoring and control to overcome this tendency and to discard a response. Together, these results suggest that when subjective recollection is salient and important for current goals, activation in the anterior insula will signal this during memory retrieval and, together with dACC and lateral PFC, will recapitulate this signal when the output of memory retrieval is evaluated as the basis for decision making.

This study is limited in that it does not include a direct manipulation of decision demands or relevance of specific task features for decision making. Recent research has underscored contributions of mechanisms signaling decision value (e.g., *Vaidya and Badre, 2020*) or supporting adaptive changes in decision criteria as the task progresses (e.g., *Scimeca et al., 2016*), but these factors are

not accounted for in the present research. Future studies manipulating incentive structures are needed to better understand the modulatory effects of metacognitive processes for decision making based on memory. We additionally note that the order of the memory and decision phases was not counterbalanced in the present experiment to preserve a naturalistic course of events, such that people render memory responses first and then make decisions about them. However, the manipulation of the order in future research would elucidate the potential effect of committing to a memory decision on subjective evaluations given demonstrated effects of choice on subsequent retention (e. g., *Murty et al., 2019*). It should also be noted that decision making in the present study was supported via a social motivation, namely the motivation to do well and/or avoid embarrassment due to poor performance. More research is needed to extend the effects of subjective recollection to decision making with financial incentives or other forms of decisions. However, the literature suggests that social and monetary rewards recruit similar neural circuits (*Ruff and Fehr, 2014*). Moreover, as the present study focused on retrospective metacognitive judgments, future research should also elucidate the degree to which prospective metacognitive judgments, such as judgments of learning, play a similar role for future decision making.

Finally, we found comparable retrieval success effects in the hippocampus between the A-A' and the A-B' conditions. These results are consistent with a previous study by *Richter et al., 2016* who found that hippocampus activation distinguished between successful vs. unsuccessful retrieval but did not vary with memory precision or subjective vividness. These results tentatively suggest that while hippocampus plays a key role for episodic memory, it is not the main driver of the subjective experience of recollection and is less dependent on the specific context in which retrieval takes place.

Taken together, the present study investigated the neural basis of subjective and objective recollection using an experimental paradigm to dissociate these naturally intertwined aspects of episodic memory. Our results highlight the critical role of distinct PPC regions in supporting memory accuracy via the identification of diagnostic feature information associated with the SPL. In contrast, activation in the retrosplenia cortex and the precuneus was associated with subjective recollection when participants retrieved item-relevant information and they did not have to compare opposing mnemonic signals regarding their diagnostic value. In line with the nature of subjective recollection judgments as metacognitive assessments that guide upcoming decision making, insular, cingulate, and lateral frontal regions signal saliency of the retrieved information at retrieval. This will then be carried over to later decision stages where these task-relevant signals are recapitulated and evaluated in the context of concurrent goals. These results provide insights into the neural mechanisms supporting our phenomenological experience of remembering and its assessment in context of our goals. Our findings also open up vistas toward a better understanding of how these mechanisms develop across the life span.

## Materials and methods

### Participants

Thirty healthy right-handed volunteers (18–25 years, M = 21.0, SD = 1.7, 15 females) with no history of or current neurological or psychiatric illness participated in the study after signing informed consent approved by the Institutional Review Board of the University of California, Davis (protocol #217322). One additional participant took part in the experiment but was excluded from further analyses due to an incidental finding. The sample size was determined based on the moderate effect sizes for the dissociation between objective and subjective memory measures observed in each of the four behavioral experiments included in *Hembacher and Ghetti, 2017* and corresponds to the largest sample employed in those behavioral experiments. One participant failed to comply with instructions, resulting in a final sample of 29 participants included in the present analyses.

### Stimuli

Stimuli included pairs of 342 color images of familiar everyday objects. These stimuli depicted very similar but distinct versions of the same object selected from *Yassa et al., 2011* and from the nternet.

## Experimental design

During the encoding task, participants were presented with 342 unique items across three runs (114 items per run). The selection of the individual item from each pair for encoding, allocation of the item to a specific block, and the order of block presentation were randomized across participants. Participants viewed each object for 1500 ms while performing an indoor–outdoor judgment. The placement of the response options (left and right index fingers) was randomized across participants. A jittered fixation cross (500–6500 ms) was presented after every image. Encoding runs were performed in the scanner but were not analyzed for the present research questions.

The retrieval task included a total of 228 2AFC recognition trials distributed across three runs with 76 trials per run. In the 2AFC recognition task, two images, a target and a lure, were presented to the left and right of the center of the screen for 3000 ms. The placement of the target was randomized across trials so that it was equally likely to occur on the left or the right for each participant. In each run, there were a total of 38 A-A' trials in which the lure depicted a different version of the same object (*Figure 1B*) and 38 A-B' trials in which the lure depicted a different version of a studied object that was not concurrently presented on the screen (and was not used as a target in other trials). Thus, all lures resembled studied items, but the target was sometimes presented beside a perceptually similar and familiar lure in the A-A' condition, and it was sometimes presented next to a perceptually dissimilar but familiar lure in the A-B' condition. The studied versions of the stimuli used as lures in the A-B' condition were not included as targets, resulting in two-thirds of all studied items, or 228 trials, that were tested as targets. Targets and lure assignments to conditions were randomized across participants.

The retrieval task included a memory phase and a decision phase (*Figure 1B*). During the memory phase, participants were instructed to select the target image by using their left hand if they thought it was the image on the left and their right hand if they thought it was the image on the right. On each side, they could select 'remember' (R) if they vividly recollected the image from the encoding phase with specific details (index finger), 'very familiar' (VF) if they knew they had previously seen the item but could not recall specific details about its presentation (middle finger), or 'somewhat familiar' (SF) if the image was familiar but to a lesser degree (ring finger). Participants were asked to use all response options (see Appendix for exact instructions to participants). After the retrieval phase, a jittered period of 500–6500 ms (in 2000 ms steps) preceded the decision phase. For longer jitters of 2500, 4500, and 6500 ms duration (ca. 50% of all trials), a number was presented on the screen every 2000 ms and participants had to press the corresponding number. This active baseline was included to prevent participants from actively thinking about their previously submitted memory response during jittering, while being undemanding enough to avoid forgetting of the preceding answer or interference with the subsequent decision phase.

During the decision phase, the two images appeared again at the same locations as during retrieval in the memory phase. Participants were asked to decide whether they wanted to select their response and count it toward their final score. A picture of a treasure chest and a trash can were placed on the left and right sides in the lower half of the screen. If they decided to select their memory, participants were asked to press the button corresponding to the treasure chest, and if they chose to discard their answer they were to press the button corresponding to the trash can (see Appendix for exact instructions to participants). The position of the treasure chest and the trash can on the screen was counterbalanced across participants. The duration of the decision phase was 1500 ms, followed by a jittered fixation period of 500–6500 ms. Participants used their pinkies to respond, thereby preventing any motor response overlap across all responses and phases of the experiment. Participants were instructed to select about 50% of the trials. To motivate participants, they were told that their final score would be ranked in comparison to all other students who had participated in the study, and their ranking would be displayed on the screen at the end of the task in the presence of the experimenter. We did not mention anything about subjective judgments or make any connections between participants' decision to select a response and their subjective judgment during the memory phase. Thus, participants believed that their score was calculated based purely on the objective memory accuracy of the items selected to be placed in the treasure chest. At the end of the task, participants were debriefed that the ranking did not exist.

## Behavioral analyses

All analyses were conducted in R (*R Development Core Team, 2020*) using RStudio (*Team R Studio, 2020*). ANOVAs were performed using the ezANOVA function in the package ez (*Lawrence, 2016*). We used mixed-effects models to examine the effects of experimental condition on accuracy, subjective recollection, and decision making. Models were implemented using the lme4 package (*Bates et al., 2014*). Mixed-effects logistic regressions were fit to single-trial data, with intercepts varying by participant. P-values were derived using lmerTest via Satterthwaite's degrees of freedom (DOF) method (*Kuznetsova et al., 2017*). Data were visualized using raincloud plots (*Allen et al., 2019*) implemented in the ggplot package (*Wickham, 2016*). To examine condition differences in evidence accumulation, we utilized drift diffusion models. The software *fast-dm* (*Voss et al., 2015*) was used to estimate the three parameters: drift rate (v), threshold separation (a), and non-decision time parameter (t0). The starting point bias was set to 0.5 (i.e., in the middle between the two thresholds); all other parameters were set to 0. We estimated v and a separately for each individual's A-A' and A-B' conditions, and t0 was estimated to be equal across conditions.

## fMRI acquisition

Images were acquired on Siemens magnetom Skyra 3T scanner (Siemens Medical AG, Erlangen, Germany) using a 32-channel head coil. Each block of the 2AFC recognition test was scanned using a gradient EPI sequence (TR = 1500 ms; TE = 24.2 ms; FOV = 216×216×138; voxel size = 3 mm isotropic). A high-resolution MPRAGE scan was obtained for co-registration of the functional images (TR = 2500 ms; TE = 3.23 ms; TI = 1100 ms; flip angle = 7 deg; FOV = 226×226×180; voxel size = 0.7 mm isotropic).

## fMRI data processing

Preprocessing was performed using FEAT in FSL 6.0.1 (https://fsl.fmrib.ox.ac.uk/fsl/fslwiki, *Woolrich et al., 2004*). The first four scans of each run were discarded to account for field inhomogeneities. Preprocessing included non-brain tissue removal, slice time and motion correction, and spatial smoothing using an 8 mm full-width half-maximum Gaussian filter. A prewhitening technique was used to account for the intrinsic temporal autocorrelation of BOLD imaging. Low-frequency artifacts were removed by applying a high-pass temporal filter (Gaussian-weighted straight-line fitting, sigma = 50 s). Registration to each participant's structural image using a boundary-based registration algorithm (*Greve and Fischl, 2009*) and to the MNI template (12 DOF) was carried out using FLIRT (*Jenkinson and Smith, 2001*).

First, to examine differences associated with the A-A' and the A-B' conditions, individual time series were modeled separately for the memory and decision phases of each trial with regressors for each condition (GLM1). Second, to examine differences between conditions specifically associated with remember and familiar responses, we modeled individual time series with separate regressors for remember and familiar responses (collapsing across VF and SF responses) in each condition (i. e., A-A' remember, A-A' familiar, A-B' remember, A-B' familiar) and during each of the memory and decision phases (GLM2). Finally, to investigate the neural signatures associated with decision making based on states of subjective recollection, in GLM3 we modeled individual time series with separate regressors for choosing to select an answer or to discard it (A-A' select, A-A' discard, A-B' select, A-B' discard) during each of the memory and decision phases.

Regressors in each model were generated by convolving the impulse function related to the onsets of events of interest with a double-gamma hemodynamic response function and were modeled with the response time of each individual trial. Motion correction parameters estimated from the realignment procedure were entered as covariates of no interest in all GLMs. In each GLM, contrast images were computed for each run per subject, spatially normalized, transformed into MNI standard space and submitted to a within-subject fixed-effects analysis across runs. Higher-level analysis across subjects was carried out using a mixed-effects model in FSL (FLAME, *Woolrich et al., 2004*). Whole-brain images were thresholded at Z > 3.1, cluster-corrected at p<0.05. ROI analyses were corrected for multiple comparisons within a contrast using a false discovery rate correction (labeled $p_{adj}$).

## Additional information

### Funding

| Funder | Grant reference number | Author |
|---|---|---|
| James S. McDonnell Foundation | Scholar Award | Simona Ghetti |
| German Research Foundation | FA 1196/1-1 | Yana Fandakova |

The funders had no role in study design, data collection and interpretation, or the decision to submit the work for publication.

### Author contributions

Yana Fandakova, Conceptualization, Data curation, Formal analysis, Investigation, Writing - original draft, Writing - review and editing; Elliott G Johnson, Investigation, Writing - review and editing; Simona Ghetti, Conceptualization, Supervision, Funding acquisition, Investigation, Methodology, Writing - original draft, Writing - review and editing

### Author ORCIDs

Yana Fandakova https://orcid.org/0000-0003-3747-0359
Simona Ghetti https://orcid.org/0000-0001-8282-0616

### Ethics

Human subjects: This research was approved by the Institutional Review Board at the University of California, Davis after signing informed consent received.(protocol #217322).

### Decision letter and Author response

Decision letter https://doi.org/10.7554/eLife.62520.sa1
Author response https://doi.org/10.7554/eLife.62520.sa2

## Additional files

### Supplementary files

• Transparent reporting form

### Data availability

Data generated or analysed during this study are included in the manuscript and supporting files. Source data files have been provided for Figures 2, 3 and 4.

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
