## [Decision Letter]

**Acceptance summary:**

This is an interesting and timely study exploring the relationship between objective and subjective indices of recollection to provide novel insights into the mechanisms underlying memory-guided decision-making. Using an innovative experimental paradigm comprising a memory phase and decision phase, the authors provide an elegant behavioural dissociation between two conditions; A-A' condition in which the diagnostic features of stimuli are prioritised thus promoting higher levels of objective accuracy, versus the A-B' condition in which a global appraisal of the target stimulus instantiates a stronger sense of subjective recollection. Results suggest that participants' behaviour is derived from subjective global appraisal, rather than fine-grained consideration of objective features.

**Decision letter after peer review:**

Thank you for submitting your article "Distinct Neural Mechanisms Underlie Subjective and Objective Recollection and Guide Memory-based Decision Making" for consideration by *eLife*. Your article has been reviewed by three peer reviewers, one of whom is a member of our Board of Reviewing Editors and the evaluation has been overseen by Timothy Behrens as the Senior Editor. The reviewers have opted to remain anonymous.

The reviewers have discussed the reviews with one another and the Reviewing Editor has drafted this decision to help you prepare a revised submission.

Summary:

This fMRI study of young healthy adults explores potential dissociations between subjective recollection and objective episodic memory accuracy within the posterior parietal cortex and PFC. Using a 2AFC object-memory task, the authors reveal clear behavioural dissociations between two conditions; A-A' condition in which the diagnostic features of stimuli are prioritised thus promoting higher levels of objective accuracy, versus the A-B' condition in which a global appraisal of the target stimulus instantiates a stronger sense of subjective recollection. The study further explores metacognitive appraisals of this process, by having participants make judgements about keeping or discarding recollection trials towards an overall score. Taken together, the behavioural and neural dissociations are interesting and timely, and this study should make a useful contribution to the literature.

Essential revisions:

1) The authors need to provide a much more thorough overview of the extant literature. For example, several studies exist that have already examined this dissociation, some cited here (Richter et al., 2016), others not cited or discussed in this way (Duarte, Henson, Graham, Cerebral Cortex 2008; Mark and Rugg, 1998) as well as PPC lesion studies (Davidson et al., Neuropsychologia 2008; Hower et al., Neuropsychologia 2014; Simons et al., 2010; Ciaramelli et al., Cortex 2017). The authors need to clearly articulate how their study builds on and extends previous work, as well as delineating precisely the novel contribution that their study makes.

2) Related to the above point, a substantial body of work exploring the contribution of posterior parietal regions to episodic recollection has not been discussed (e.g. Kuhl and Chun, J.Neuroscience 2014; Favila et al., J. Neuroscience 2018; Frithsen and Miller., Neuropsychologia 2014; Ramanan S. et al., The Neuroscientist, 2018). It would be important to integrate this prior work to appreciate how the current results fit within the broader memory literature.

3) It is not clear how the distinction between subjective and objective recollection maps on to the distinction made between "local" and "global" processing. The authors should clearly explain what they mean by "global assessment" (provide a definition) and why exactly this process might give rise to a stronger sense of subjective recollection (e.g. as proposed in the Introduction). Similarly, the authors should clarify why this "global assessment" would also come alongside a situation where "salient retrieval output will be more likely to serve as a basis for response".

4) While it is acceptable to have an active baseline to reduce the probability of negative BOLD memory effects, due to greater activity at rest, it seems highly unlikely that the baseline task ("press a number") would prevent participants from remembering their previous memory decision just a few seconds earlier. Given that the decision phase task asks participants to keep or trash their prior memory decision, it is not clear how this would be an effective distractor task, as subjects knew they needed to make this decision based on their first memory choice. Furthermore, it is stated in the decision phase results that the authors wanted to determine if patterns of activity in the memory phase persisted in the decision phase, which of course, many did. While the decision phase may provide some behavioural support for the authors' predictions about the two task conditions, it is not clear what the decision vs. memory phase neural comparison is intended to show. Presumably, the authors could compare BOLD responses during the memory phase for select and discard trials and see similar results, though peaks and significance values may vary somewhat.

5) The precise instructions and conditions of the decision phase of the experiment warrant further explanation. Under what circumstances would a participant “trash” their responses? What were the specific instructions provided to participants? Were participants instructed explicitly on how their score was calculated? Did they think their score simply reflected the correct versus incorrect retrievals? It is not clear from the manuscript whether participants thought their score was calculated based purely on objective retrieval accuracy or whether it also related to their "remember vs. very familiar vs. somewhat familiar" judgements. This is a subtle point but one with serious implications for the interpretation of the behavioural and neural data.

6) Related to the neural predictions on the decision phase, there is no discussion of the differences between retrospective and prospective metacognitive monitoring within the PFC (e.g. Fleming and Dolan, Phil. Trans. Royal Society. 2012). Metacognitive monitoring is treated as a unitary construct here.

7) Please detail how the sample size was determined – was a power analysis conducted? Was this sample a convenience sample or determined based on available resources? What was the stopping rule for data collection? Additionally, was the study pre-registered? Which hypotheses and tests were a-priori, and which were conducted after the data had been examined? Ideally, this should be specified throughout.

8) Further details are required regarding the data analysis strategy. For example, for the mixed-effects modelling approach, was there a reason why only intercepts were allowed to vary by participant but not random slopes? There are good theoretical and empirical reasons to expect that the coefficients for the key effects would also vary by participant, not only the intercepts. Some justification of the modelling decisions is warranted here.

9) Similarly, the reader would benefit from more detail about where the reported results are drawn from – in the mixed effects framework, for instance, where are the p values derived? Did you use a package like "lmerTest" with Satterthwaite's method, or did you do some kind of model selection (comparing between models with/ without focal variables)? As far as I know, lme4 does not, by default, provide p-values.

10) The plots lack the overlaid raw data, making inferences difficult. The authors should overlay the raw datapoints on top of the bar charts so that readers can see for themselves what the distributions look like. Similarly, there are no plotted logistic regression functions, meaning the plots for the behavioural data do not match up with the analytical tools used for the inferential statistics and reported in text – perhaps this needs at least to be explained in the figure legend?

11) A point worthy of discussion is the possibility that the motivation to do well on this task may not necessarily be financial, but could have been socially driven, in that participants' scores were displayed. Thus, motivation to perform well may not reflect "reward" as per previous incentive-compatible confidence studies, but potentially both social reward and punishment (e.g. embarrassment at poor performance). Aspects of the Abstract and Discussion should be re-phrased because the decision task was not incentivized with tangible rewards – instead, it could also contain the motivation to avoid a penalty like embarrassment.

12) I would like to see some consideration of the laterality of the parietal regions that were recruited during the memory phase, given that it was the right supramarginal gyrus that emerged in the analyses. For example, I was surprised that the angular gyrus did not emerge as a key region during the memory phase and I wonder if the authors could comment on the lack of AG involvement in the current study (e.g. see work by Preston Thakral; Siddharth Ramanan, Heidi Bonnici). It would help the reader to place some of these findings in context and comment on how the paradigms etc. potentially give rise to these differences across studies.

---

## [Author Response]

Essential revisions:1) The authors need to provide a much more thorough overview of the extant literature. For example, several studies exist that have already examined this dissociation, some cited here (Richter et al., 2016), others not cited or discussed in this way (Duarte, Henson, Graham, Cerebral Cortex 2008; Mark and Rugg, 1998) as well as PPC lesion studies (Davidson et al., Neuropsychologia 2008; Hower et al., Neuropsychologia 2014; Simons et al., 2010; Ciaramelli et al., Cortex 2017). The authors need to clearly articulate how their study builds on and extends previous work, as well as delineating precisely the novel contribution that their study makes.

We thank the reviewer for encouraging us to consider these papers. We have now included them in the Introduction and Discussion of the revised manuscript, and we have clarified the novel contribution of our work. Specifically, we emphasize that previous evidence regarding the neural dissociation between subjective and objective recollection comes primarily from populations in which either one or the other aspect of recollection is altered (e.g., older adults or patients with PPC damage; Duarte et al., 2008; Simons et al., 2010). In contrast, dissociating these processes in typically-developing adults has proved quite difficult because objective and subjective recollection measures are typically highly correlated. Indeed, even in studies where task instructions are designed to trigger these different types of recollection, the two aspects continue to be highly correlated (i.e., a successful source judgment will typically be associated with reports of subjective recollection). This limits our ability to isolate correlates of one aspect of recollection even when the other is statistically controlled for. Overall, it has been difficult to fully evaluate the distinction between objective and subjective recollection in young, healthy brains. Our experiment was designed to dissociate subjective and objective recollection allowing us to directly examine the underlying neural correlates.

2) Related to the above point, a substantial body of work exploring the contribution of posterior parietal regions to episodic recollection has not been discussed (e.g. Kuhl and Chun, J.Neuroscience 2014; Favila et al., J. Neuroscience 2018; Frithsen and Miller., Neuropsychologia 2014; Ramanan S. et al., The Neuroscientist, 2018). It would be important to integrate this prior work to appreciate how the current results fit within the broader memory literature.

We have now added a discussion of these papers (Introduction and Discussion). We think these papers have helped us to integrate the present results in the broader literature and thank the reviewer for the helpful suggestions.

3) It is not clear how the distinction between subjective and objective recollection maps on to the distinction made between "local" and "global" processing. The authors should clearly explain what they mean by "global assessment" (provide a definition) and why exactly this process might give rise to a stronger sense of subjective recollection (e.g. as proposed in the Introduction). Similarly, the authors should clarify why this "global assessment" would also come alongside a situation where "salient retrieval output will be more likely to serve as a basis for response".

We thank the reviewer for making us aware that this discussion lacked clarity. We have now revised the Introduction to define more clearly what we mean by local vs. global processing. In addition, to be more precise we now use describe the processing as specific rather than local:

“However, participants are consistently less likely to claim subjective recollection in the A-A’ condition, likely because it forces them to identify the most diagnostic feature from the comparison between the two perceptually similar probes. […] Instead it encourages participants to assess their retrieval of the probe as a whole, resulting in more errors, but also in a stronger sense of subjective recollection.“

Thus, in line with previous research using this paradigm, we expect a retrieval context promoting more specific retrieval to be associated with a greater likelihood for objective recollection at the cost of reduced subjective recollection. In contrast, a retrieval context promoting global retrieval is expected to increase subjective recollection and potentially reduce objective performance, especially when the to-be-remembered information/feature is highly specific.

We also clarified the link between more global processing favoring subjective recollection, and saliency:

“From this perspective, we speculate that the enhanced anterior insular activation in the A-B’ condition (and in other tasks emphasizing judgments of subjective recollection) reflects the fact that the most goal-salient aspect of the task is to identify which items are subjectively recollected.”

4) While it is acceptable to have an active baseline to reduce the probability of negative BOLD memory effects, due to greater activity at rest, it seems highly unlikely that the baseline task ("press a number") would prevent participants from remembering their previous memory decision just a few seconds earlier. Given that the decision phase task asks participants to keep or trash their prior memory decision, it is not clear how this would be an effective distractor task, as subjects knew they needed to make this decision based on their first memory choice. Furthermore, it is stated in the decision phase results that the authors wanted to determine if patterns of activity in the memory phase persisted in the decision phase, which of course, many did. While the decision phase may provide some behavioural support for the authors' predictions about the two task conditions, it is not clear what the decision vs. memory phase neural comparison is intended to show. Presumably, the authors could compare BOLD responses during the memory phase for select and discard trials and see similar results, though peaks and significance values may vary somewhat.

We were interested in attempting to isolate the neural substrates associated with behavior during the memory phase from those associated with behavior during the decision phase. Thus, we were confronted with the question of how to best fill the jittering time between the memory and decision probes in a way that maximized our chances to address our question. Leaving the time empty would probably have introduced confounds as different participants may have engaged in continued retrieval attempts, rehearsal or preparation for decision making at different times. Including a demanding task would have risked eliminating any of such potential sources of variability, but also interfering with participants’ ability to keep their memory decision in mind, thereby introducing a new demand for memory retrieval during the decision phase.

Our choice of an active baseline task was thus motivated by our attempt to systematically limit participants’ further engagement in retrieval processes or preparation for the upcoming decision, while avoiding an overly taxing task that could result in forgetting the answers submitted during the memory phase. With the active baseline, we thus sought to increase comparability to the previous behavioral studies with this paradigm (Hembacher and Ghetti, 2016), in which delays between memory and selection were shorter than the jittered fixation periods necessary in the context of the present event-related fMRI design. While we agree with the reviewer that it would be surprising to find completely non-overlapping activation for the contrast of select vs. discard responses performed at the memory vs. decision phase, we assumed that activation during the memory phase would primarily reflect recollection-based processes. Thus, we expected that analyzing the decision to select or discard a response during the decision phase would allow us to better isolate the neural substrates associated with these decisions.

We performed the contrast of select vs. discard responses at the memory phase as well. As can be seen in Author response image 1, the activations are largely overlapping, especially for the discard decisions. However, for select responses, activation during the memory phase is much more widespread and includes clusters observed in association with remember responses, reflecting the fact that decisions were tightly coupled with subjective recollection.

**Author response image 1. sa2fig1:** Select > discard (in yellow/red) and discard > select (in blue) during the memory phase (A) and during the decision phase (B; reported in main manuscript). C. Overlay of select > discard from memory phase (in dark red) and select > discard from the decision phase (in yellow/red). D. Overlay of discard > select from memory phase (in dark blue) and discard > select from the decision phase (in light blue).

5) The precise instructions and conditions of the decision phase of the experiment warrant further explanation. Under what circumstances would a participant “trash” their responses? What were the specific instructions provided to participants? Were participants instructed explicitly on how their score was calculated? Did they think their score simply reflected the correct versus incorrect retrievals? It is not clear from the manuscript whether participants thought their score was calculated based purely on objective retrieval accuracy or whether it also related to their "remember vs. very familiar vs. somewhat familiar" judgements. This is a subtle point but one with serious implications for the interpretation of the behavioural and neural data.

We thank the reviewer for the opportunity to clarify our procedure. Participants were informed that at the end of the experiment their performance would be compared to that of their peers who completed the experiment. Specifically, we informed participants that to compute the score used for the comparison we would only use the answers selected for the treasure chest, thereby discarding the answers in the trash bin. Thus, if a participant did not want their answer to be counted towards their final score, they had the opportunity to discard it by using the “trash” option. Of note, we did not mention anything about subjective states and/or the relation between their subjective judgment of “remember,” “very familiar,” or “somewhat familiar,” and their upcoming decision. Thus, participants believed that their score was calculated based purely on the objective retrieval accuracy for the treasured items. We have now clarified this point in the manuscript. After completing the task, participants were debriefed and we informed them that we would not be comparing their performance to others’. We have now provided the exact instructions participants received in the Appendix.

6) Related to the neural predictions on the decision phase, there is no discussion of the differences between retrospective and prospective metacognitive monitoring within the PFC (e.g. Fleming and Dolan, Phil. Trans. Royal Society. 2012). Metacognitive monitoring is treated as a unitary construct here.

We thank the reviewer for alerting us to a lack of clarity on our part. We have now clarified that we focus on retrospective metacognitive monitoring in the present study in the Introduction and the Discussion. Additionally, we have now noted in the Discussion that future research is needed to examine whether and how prospective metacognitive judgments, such as judgment of learning, guide future decisions:

“Moreover, as the present study focused on retrospective metacognitive judgments, future research should also elucidate the degree to which prospective metacognitive judgments, such as judgments of learning, play a similar role for future decision making.”

7) Please detail how the sample size was determined – was a power analysis conducted? Was this sample a convenience sample or determined based on available resources? What was the stopping rule for data collection? Additionally, was the study pre-registered? Which hypotheses and tests were a-priori, and which were conducted after the data had been examined? Ideally, this should be specified throughout.

We did not conduct a formal power analysis to determine the sample for the present experiment. We selected the sample size for this experiment (N = 30) based on the moderate effect sizes for the dissociation between objective and subjective memory measures observed across each of four behavioral experiments included in Hembacher and Ghetti, 2016. The current sample size corresponds to the largest sample employed in one of those behavioral experiments. The stopping rule for data collection was to achieve the sample size of N = 30. Due to an incidental finding in one participant, their data were replaced (and not used for analyses), resulting in N = 31 participating adults. In the process of data analyses, it became clear that one participant had not followed the instructions and was excluded from analyses. However, their data were not replaced, resulting in an effective sample of N = 29. We have now included this information in the revised manuscript.

The present study was not pre-registered. We had a-priori expectations regarding the dissociation between subjective and objective recollection (based on previous work with this paradigm, Hembacher and Ghetti, 2016; see also Selmeczy, Kazemi and Ghetti, 2021), the condition dissociation in the parietal cortex (based on close alignment of the experimental conditions with bottom-up vs. top-down attention to memory, e.g., Cabeza et al., 2008) as well as PFC, insula and dACC involvement in the decision to select or discard responses (based on our own developmental metamemory work, e.g., Fandakova et al., 2017, 2018).

8) Further details are required regarding the data analysis strategy. For example, for the mixed-effects modelling approach, was there a reason why only intercepts were allowed to vary by participant but not random slopes? There are good theoretical and empirical reasons to expect that the coefficients for the key effects would also vary by participant, not only the intercepts. Some justification of the modelling decisions is warranted here.

We included a random factor for the intercept to account for overall individual variability in performance in our analysis. We did not include random slope effects in the estimated logistic regressions because we did not have strong expectations about individual variability in the effect of the experimental manipulation and our previous studies had shown showed a moderate-to large effect sizes for our manipulation. In our sample, 83% (24 out of 29 participants) showed the expected effect of the manipulation (i.e., higher accuracy in the AA’ condition accompanied by higher subjective recollection in the A-B’ condition). However, we agree with the reviewer that one might expect individual variability in the extent of the manipulation effects. Our results are fully replicated if we also include random slopes for the condition effects. We elected to keep the current analysis because it is simpler and more accessible, but we would be happy to include additional analyses if the reviewers and Editors deem this helpful or necessary.

9) Similarly, the reader would benefit from more detail about where the reported results are drawn from – in the mixed effects framework, for instance, where are the p values derived? Did you use a package like "lmerTest" with Satterthwaite's method, or did you do some kind of model selection (comparing between models with/ without focal variables)? As far as I know, lme4 does not, by default, provide p-values.

We are grateful to the reviewer for pointing out that we left out this important detail in our manuscript. As anticipated by the reviewer, we used the package lmerTest with Satterthwaite’s degrees-of-freedom method. We have now added this information to the Materials and methods section of the manuscript.

10) The plots lack the overlaid raw data, making inferences difficult. The authors should overlay the raw datapoints on top of the bar charts so that readers can see for themselves what the distributions look like. Similarly, there are no plotted logistic regression functions, meaning the plots for the behavioural data do not match up with the analytical tools used for the inferential statistics and reported in text – perhaps this needs at least to be explained in the figure legend?

We have now revised all of the figures and instead of bars showing mean effects we now present raincloud plots with raw data, probability density, and key summary statistics for all effects reported in the manuscript. We decided to include the raw data instead of the estimated logistic regression functions in order to facilitate the correspondence between the regression results and the actual experimental data. Following the reviewer’s suggestion, we have now clarified this fact in the legend of Figure 2 and also provide the plotted logistic regression functions in Figure 2—figure supplement 1.

11) A point worthy of discussion is the possibility that the motivation to do well on this task may not necessarily be financial, but could have been socially driven, in that participants' scores were displayed. Thus, motivation to perform well may not reflect "reward" as per previous incentive-compatible confidence studies, but potentially both social reward and punishment (e.g. embarrassment at poor performance). Aspects of the Abstract and Discussion should be re-phrased because the decision task was not incentivized with tangible rewards – instead, it could also contain the motivation to avoid a penalty like embarrassment.

We have now clarified that the reward in the present study was not financial, but social – the incentive was to do well in comparison to peers. We have also indicated that future research is needed to confirm that subjective recollection similarly guides decision making when financial gains or losses are expected (Discussion). Indeed, there are hints in the literature suggesting that the neural networks supporting social decision making are similar to the neural circuits identified in non-social situations (e.g., Izuma et al., 2008; Lin et al., 2012; Ruff and Fehr, 2014).

12) I would like to see some consideration of the laterality of the parietal regions that were recruited during the memory phase, given that it was the right supramarginal gyrus that emerged in the analyses. For example, I was surprised that the angular gyrus did not emerge as a key region during the memory phase and I wonder if the authors could comment on the lack of AG involvement in the current study (e.g. see work by Preston Thakral; Siddharth Ramanan, Heidi Bonnici). It would help the reader to place some of these findings in context and comment on how the paradigms etc. potentially give rise to these differences across studies.

We thank the reviewer for encouraging us to position our results better and provide additional clarification. We now report that the present paradigm did produce reliable activity in left AG during the memory phase, as expected based on previous research. As depicted in Figure 3—figure supplement 1, the contrast generated by subjective judgments (Remember > Familiar) produced reliable activation in left angular gyrus in both the A-A’ (red) and the A-B’ (blue) conditions, and there were no differences between conditions in this region. Thus, the results involving this general contrast during the memory phase, irrespective of condition, are fully consistent with the literature. Despite this general effect in the angular gyrus, the condition differences found in the present study involve other parietal subregions. The results regarding the SPL are consistent with previous studies showing that memory goals bias feature representations towards relevant information in dorsal PPC, but not in ventral PPC (Favila et al., 2018). Based on previous studies, one might expect to find the increased subjective recollection effects in the angular gyrus, but instead we found them in the precuneus. While these results corroborate the findings of Richter et al., 2016, they also help to clarify the role of the angular gyrus – previous research implicating this area in subjective recollection has not accounted for the fact that the precision of retrieved memories and the subjective experience of recollection are highly correlated (see Ramanan et al., 2018). By dissociating accuracy from recollection in the present study, we showed that precuneus activity varied with the subjective experience of recollection. We have now included these results in Figure 3—figure supplement 1 and refer to them in the Results section.